# Low incidence of HCC in chronic hepatitis C patients with pretreatment liver stiffness measurements below 17.5 kilopascal who achieve SVR following DAAs

Jacob Søholm[1,2,3]*, Janne Fuglsang Hansen[1], Belinda Mössner[1,2], Birgit Thorup Røge[4], Alex Lauersen[5], Jesper Bach Hansen[6], Nina Weis[7,8], Toke Seierøe Barfod[9], Suzanne Lunding[10], Anne Øvrehus[11], Rajesh Mohey[12], Peter Thielsen[13], Peer Brehm Christensen[1,2]

1 Department of Infectious Diseases, Odense University Hospital, Odense, Denmark, 2 OPEN, Odense Patient data Explorative Network, Odense University Hospital, Odense, Denmark, 3 Clinical Institute, University of Southern Denmark, Odense, Denmark, 4 Department of Medicine, Lillebaelt Hospital, Kolding, Denmark, 5 Department of Infectious Diseases, Aarhus University Hospital, Skejby, Aarhus, Denmark, 6 Department of Gastroenterology, Aalborg University Hospital, Aalborg, Denmark, 7 Department of Infectious Diseases, Copenhagen University Hospital, Hvidovre, Denmark, 8 Department of Clinical Medicine, Faculty of Health and Medical Sciences, University of Copenhagen, Copenhagen, Denmark, 9 Department of Medicine, Zealand University Hospital, Roskilde, Denmark, 10 Department of Pulmonary and Infectious Diseases, North Zealand University Hospital, Hilleroed, Denmark, 11 Department of infectious Diseases, Copenhagen University Hospital Rigshospitalet, Copenhagen, Denmark, 12 Infectious Disease Unit, Department of Medicine, Herning Hospital, Herning, Denmark, 13 Department of Gastroenterology, Copenhagen University Hospital, Herlev, Denmark

* Jacob.Soeholm@rsyd.dk

**Data Availability Statement:** Data cannot be shared publicly because of the Data Protection Laws in Denmark. Data are available for other

## Abstract

### Background and aims

To evaluate the ability of pretreatment liver stiffness measurements (pLSM) to predict hepatocellular carcinoma (HCC), incident decompensation and all-cause mortality in chronic hepatitis C (CHC) patients who achieved sustained virological response (SVR) after treatment with direct-acting antivirals (DAAs).

### Methods

773 CHC patients with SVR after DAA treatment and no prior liver complications were identified retrospectively. Optimized cut-off of 17.5 kPa for incident HCC was selected by maximum Youden's index. Patients were grouped by pLSM: <10 kPa [reference], 10–17.4 kPa and ≥17.5 kPa. Primary outcomes were incident hepatocellular carcinoma and secondary outcomes were incident decompensated cirrhosis and all-cause mortality, analyzed using cox-regression.

### Results

Median follow-up was 36 months and 43.5% (336) had cirrhosis (LSM>12.5 kPa). The median pLSM was 11.6 kPa (IQR 6.7–17.8, range 2.5–75) and pLSM of <10 kPa, 10–17.4

researchers upon approval by the Danish Patient Safety Authority (https://stps.dk/en) and the Danish Data Protection Agency (https://www.datatilsynet.dk/English/). Request for access to the data can be addressed to the Danish Patient Safety Authority (stps@stps.dk).

**Funding:** The author(s) received no specific funding for this work.

**Competing interests:** The authors have declared that no competing interests exist.

kPa and 17.5–75 kPa was seen in 41.5%, 32.2% and 26.3%. During a median follow-up time of 36 months, 11 (1.4%) developed HCC, 14 (1.5%) developed decompensated cirrhosis, and 38 (4.9%) patients died. A pLSM of 17.5 kPa identified patients with a high risk of HCC with a negative predictive value of 98.9% and incidence rate of HCC in the 17.5–75 kPa group of 1.40/100 person years compared to 0.14/100 person years and 0.12/100 person years in the 10–17.4 kPa and <10 kPa groups, $p < 0.001$.

## Conclusion

Pretreatment LSM predicts risk of HCC, decompensation and all-cause mortality in patients with SVR after DAA treatment. Patients with a pLSM <17.5 kPa and no other risk factors for chronic liver disease appear not to benefit from HCC surveillance for the first 3 years after treatment. Longer follow-up is needed to clarify if they can be safely excluded from post treatment HCC screening hereafter.

## Introduction

Treatment with direct acting antivirals (DAAs) has been shown to decrease mortality and liver related complications in patients with chronic hepatitis C (CHC) [1–7]. Current guidelines state that patients with advanced fibrosis/cirrhosis (Metavir F3-4) need continued post-treatment surveillance for hepatocellular carcinoma (HCC) every six months [8]. Given the huge number of patients with advanced fibrosis and cirrhosis who will be cured with DAAs in the coming years, it is a major research priority to identify patients who do not need to enter surveillance for liver related complications after cure [9]. People cured for hepatitis C might have other health issues and/or competing priorities that can make adherence to surveillance challenging. This is especially an issue in marginalized populations such as people who inject drugs, but limiting redundant procedures should be a priority for the health system at large [10].

A liver stiffness measurement (LSM) using vibration-controlled transient elastography (VCTE) is currently being used to identify patients with advanced liver disease and has been able to predict the prognosis of patients with chronic hepatitis [11–14]. Recently it has also been shown to be useful in predicting outcome in patients treated with CHC treated with DAAs [15–19], but most studies have focused on cirrhotic patients. Using LSM as a predicting marker allows for less contacts with health care providers before treatment initiation, as compared to biomarkers. This can be advantageous in outreach programs among marginalized populations, such as homeless people and people who inject drugs (PWID).

The primary aim of this study was to evaluate the ability of pretreatment LSM (pLSM) to predict incident decompensated cirrhosis and HCC and all-cause mortality in a cohort of patients treated with DAAs. The secondary aims were to evaluate the prognostic ability of post treatment LSM dynamics.

## Material and methods

This study was approved by the Danish Patient Safety Authority (j. nr. 3-3013-2307/1) and the Danish Data Protection Agency (j. nr. 17/29317).

### Setting

Denmark has a population of 5.8 million people [20] and a HCV prevalence of 0.4% [21]. Patients with HCV infection are treated by infectious disease specialists or hepatologists in

inpatient- or outpatient clinics. Medical care is provided free-of-charge; including treatment with DAAs. The indication for DAA treatment in Denmark has been conservative: until February 2017 LSM >12 kilopascal (kPa), hereafter it was decreased to >10 kPa and from November 2018, treatment has been offered regardless of liver fibrosis.

### Data sources

We used the unique 10-digit registration number (PIN) assigned to all Danish citizens to link individual level data from the following sources.

### The Danish Database for Hepatitis B and C (DANHEP)

We identified patients with CHC and chronic hepatitis B in the Danish Database for Chronic Hepatitis B and C (DANHEP) which is a nationwide database including all patients referred to hospital for chronic hepatitis B and C in Denmark [22].

### Liver stiffness measurements

The LSM were extracted directly from the Fibroscan software from centers that report to DANHEP. Only valid procedures with at least 10 validated measurements and an interquartile range (IQR) <30% in LSM ≥7.1 kPa were used in the study [23].

### The Danish Civil Registration System (CRS)

The CRS was established in 1968 and contains daily updated information on the vital status of all Danish citizens [24].

### The Danish National Patient Register (DNPR)

The DNPR, established in 1977, records all hospital admissions to non-psychiatric hospitals in Denmark. Data from outpatient clinics and emergency departments started in 1995. Records for each medical contact includes the dates of admission and discharge and up to 20 discharge diagnoses, coded according to the *International Classification of Diseases*, 10th revision from 1994 and onwards [25].

We used DNPR to identify heavy alcohol use, human immunodeficiency virus (HIV) status, intravenous drug use, diabetes, decompensated cirrhosis and HCC.

### Danish Registry of Causes of Death (DRCD)

DRCD contains information from all Danish death certificates issued since 1943. Computerized and validated registry information is currently available through 2017. Whenever a Danish resident dies, the attending physician must report the cause of death. Causes of death recorded during the study period were coded using ICD-10.

### The Registry of Drug Users Undergoing Treatment (RDT)

The RDT is run by the National Board of Health in Denmark and registers all individuals receiving treatment for drug addiction since 1996. It contains information on main drugs used, route of administration and whether a patient receives opioid substitution treatment or not [26].

We used the RDT to ascertain intravenous drug use (IDU).

### The Danish Cancer Registry (DCR)

The DCR contains records of all incidences of malignant neoplasms in the Danish population since 1943 [27]. It is divided into personal characteristics at date of diagnosis and tumor characteristics. We used DCR to ascertain HCC.

### The Danish Pathology Register (DPR)

The DPR contains electronic registration of all pathological specimens since 1997, when it became a legal obligation for all departments of pathology in Denmark to report to the national board of health [28]. The information consists mainly of patient data and pathology diagnoses using a Danish version of the Systematized Nomenclature of Medicine (SNOMED). The DPR was used to ascertain HCC.

### The Danish National Prescription Registry (DNPR)

The DNPR contains data on all dispensed prescriptions from Danish community pharmacies since 1994 [29]. It was used to ascertain heavy alcohol use.

### National Registry of Alcohol Treatment (NRAT)

The NRAT contains information on all public as well as private treatment for heavy alcohol consumption since 1996 [30]. Data includes information on current treatment, previous treatment, age of first alcohol ingestion and reasons for stopping treatment. The NRAT was used to ascertain heavy alcohol use.

### Study population

To be eligible for the study, patients identified as having CHC in DANHEP had to meet the following criteria: (a) age 18 years or older at inclusion, (b) a positive test for HCV antibodies and a concurrent or later positive HCV RNA (c) sustained virological response (SVR) post treatment with DAAs, defined as HCV RNA result below the lower limit of quantification at least 12 weeks after the end of treatment (d) a valid LSM < 2 years before initiation of treatment with DAA (e) no coinfection with hepatitis B virus (HBV) or HIV (f) no episode of decompensated cirrhosis or HCC prior to inclusion (g) no liver transplant prior to inclusion (h) no HCC diagnosis <180 days after index date, in order to only include patients with incident HCC after treatment initiation.

   The study included all patients fulfilling these criteria from 25. July 2012 until 24. May 2019. The index date was at the time of treatment initiation with DAAs and patients were followed until death, immigration or 24. May 2019, whichever came first.

### Definition of specific diagnoses

The definitions of diagnoses in the form of heavy alcohol use, hepatitis B, HIV, IDU, diabetes, HCC, decompensated cirrhosis and western origin are shown in S1 Appendix.

### Prognostic categories

We used cut-offs of pLSM of 10 in our study to define patients in the reference group. A cut-off of 10 kPa is used to define severe fibrosis (F3) in patients with CHC, thereby identifying patients who need continued post-treatment surveillance for HCC every six months [8] and patients with a LSM < 10 kPa, were used as the reference group. The optimized cut-off using

Youden's index for predicting incident HCC after treatment in our cohort was used for comparison. Cirrhosis was defined as having a LSM >12.5 kPa [8].

In patients with repeated LSM, performed at least 90 days after treatment initiation, we calculated the delta LSM (dLSM), defined as the difference in kPa between the last post treatment LSM and the pLSM. The dLSM was reported as ≥0 or <0. For in whom the last post treatment LSM was <10 kPa, dLSM was reported as <0 to avoid categorizing insignificant fluctuations in LSM as possible fibrosis progression.

## Outcomes

The primary outcome was incident HCC and secondary outcomes were incident decompensated cirrhosis and all-cause death.

## Statistical analysis

Person-years at risk were computed from the index date until the date of event, emigration, or 24. May 2019, whichever came first.

The significance level was set at a p-value < 0.05. Proportions and medians were compared using Pearsons Chi$^2$ and Kruskal-Wallis median test.

The cut-offs for optimized sensitivity and specificity, 90% sensitivity and specificity and positive predictive value (PPV) and negative predictive value (NPV) was found by ROC analysis and used in subsequent analyses, together with cut-off of 10 kPa and 12.5 kPa.

We computed incidence rates for all-cause mortality, HCC and decompensated cirrhosis with 95% confidence intervals (CI). Kaplan-Meier survival curves were compared using log-rank test. Cox regression was used to estimate hazard ratios for all-cause death. Competing risk regression was used to estimate subhazard ratios for HCC and decompensated cirrhosis, as death was a substantial competing risk for these two outcomes. The models included variables selected *a priori* of age, sex, Western European origin, diabetes, history of heavy alcohol use and intravenous drug use (IDU) and pLSM. Delta LSM was included in the univariate analysis but was omitted from multivariate analysis because of too few outcomes among patients with a follow-up LSM. Predictors that were associated with the outcomes with a p-value <0.0.05 were entered in a multivariate analysis.

All analyses were performed using STATA 15 IC software (Statacorp LP, College Station, TX).

## Results

Out of 1,763 patients evaluated for inclusion, 773 were included in the study (Fig 1).

The clinical characteristics of the study cohort are shown in Table 1.

Overall, the median age was 54 years (IQR 45–61, range 18–83), 63.7% were male and 84.2% were of Western origin. Diabetes was found in 10.5%, 51.2% had a registration of heavy alcohol use and 63.1% had ever injected drugs.

The median pLSM was 11.6 kPa (IQR 6.7–17.8, range 2.5–75) and using the optimized cut-off for incident HCC of 17.5 kPa (Fig 2), a pLSM of <10 kPa, 10–17.4 kPa and 17.5–75 kPa was seen in 41.5%, 32.2% and 26.3, respectively. Cirrhosis, defined as a pLSM >12.5 kPa, was seen in 336 patients (43.5%). In 98.3% (760/773) at least one LSM had been performed before the pLSM.

Compared to patients with a pLSM < 10 kPa, patients with a pLSM ≥17.5 kPa were older (56 years (IQR 49–61) vs 50 years (IQR 42–60), *p*<0.001), were more likely to be male (67.0% vs 57.6%, *p* = 0.013), have a diagnosis of diabetes (17.2% vs 4.4%, *p*<0.001) or have a registration of heavy alcohol use (62.1% vs 43.9% *p*<0.001). Genotype (GT) 1 was found in 52.3%

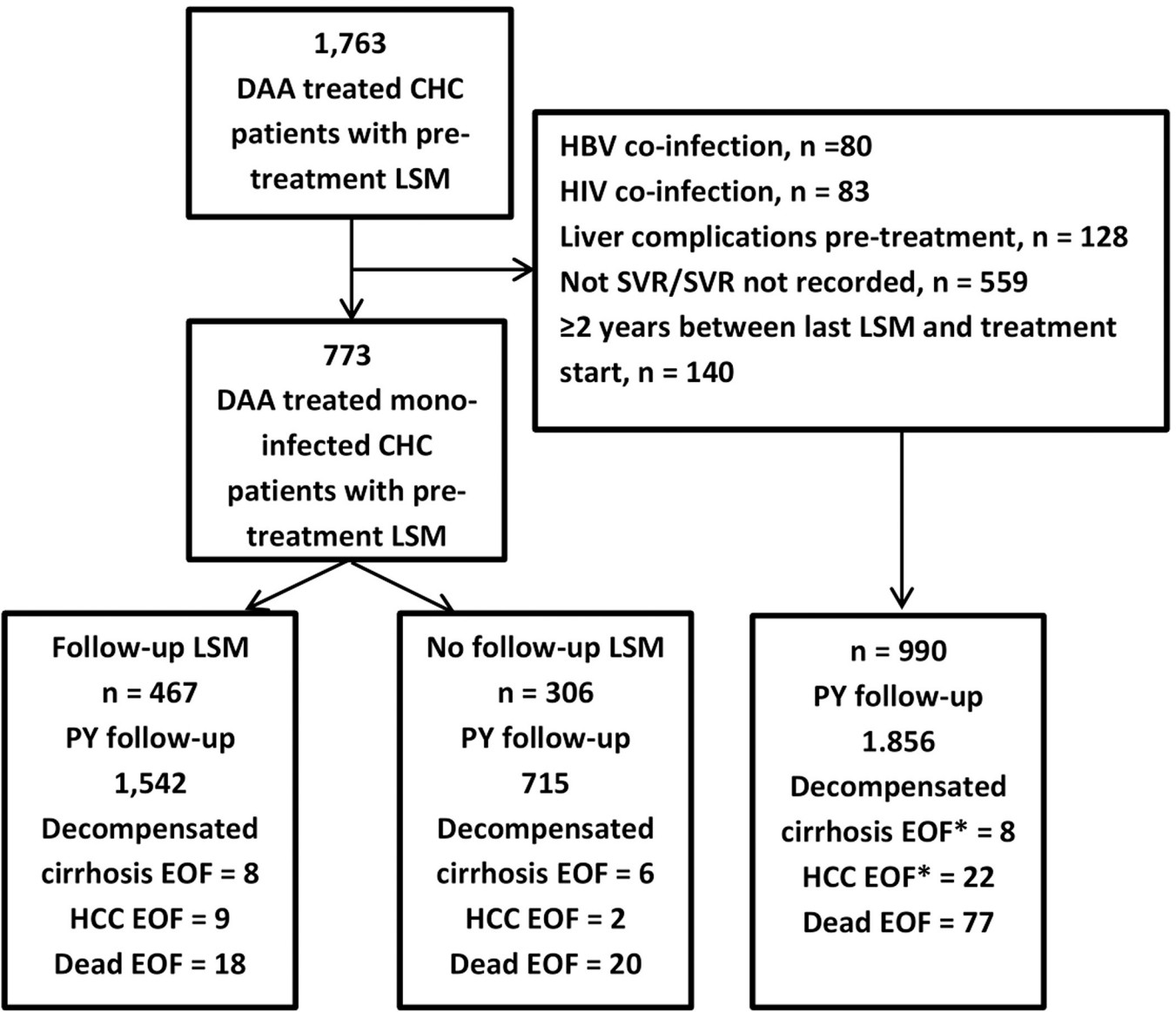

**Fig 1. Flow chart for inclusion.** Abbreviations: DAA; direct-acting antivirals, CHC; chronic hepatitis C, LSM; liver stiffness measurement, HBV; hepatitis B virus, HIV; human immunodeficiency virus, PY; person-years, EOF; end of follow-up. * First episode.

(348/666) and GT 3 in 39.2% (261/666) with 22.7% of patients with GT 1 having a pLSM ≥17.5 kPa, compared to 24.9% of patients with GT 3 (p = 0.527).

The median follow-up period was 36 months (IQR 24–47, range 6–82) and the median time from performance of pLSM to treatment initiation was 49 days (IQR 5–154).

### Follow-up liver stiffness measurements

Among 467 patients with at least one follow-up LSM, the median time from treatment initiation to last LSM was 35 months (IQR 20–53) and the median follow-up was 41 months (IQR 30–49).

Patients with a follow-up LSM were older than patients without a follow-up LSM (median age 55 vs 52 years, p = 0.019) and had a higher pLSM (12.4 vs 9.9 kPa, p<0.001) but there was no difference in sex, prevalence of diabetes, heavy alcohol use or IDU among the two groups.

**Table 1. Demographic characteristics for 773 patients with chronic hepatitis C treated with direct acting antivirals at baseline according to pretreatment liver stiffness measurement.**

| | <10 kPa | 10–17.4 kPa | 17.5–75 kPa | P | All |
|---|---|---|---|---|---|
| | **n = 321 (41.5%)** | **n = 249 (32.2%)** | **n = 203 (26.3%)** | | **n = 773** |
| **Median age, years (IQR)** | 50 (42–60) | 55 (47–61) | 56 (49–61) | **<0.001** | 54 (45–61) |
| **Male sex, n (%)** | 185 (57.6) | 171 (68.7) | 136 (67.0) | **0.013** | 492 (63.7) |
| **Western origin, n (%)** | 276 (86.0) | 204 (81.9) | 171 (84.2) | 0.420 | 651 (84.2) |
| **Diabetes, n (%)** | 14 (4.4) | 32 (12.9) | 35 (17.2) | **<0.001** | 81 (10.5) |
| **Heavy alcohol use, n (%)** | 141 (43.9) | 129 (51.8) | 126 (62.1) | **<0.001** | 396 (51.2) |
| **Intravenous drug use, n (%)** | 201 (62.6) | 153 (61.5) | 134 (66.0) | 0.588 | 488 (63.1) |
| **Median ALAT at pLSM (IQR)** | 63 (39–102) | 78 (46–134) | 97 (57–145) | **<0.001** | 75 (44–125) |
| **Median Follow-up, months (IQR)** | 32 (20–42) | 36 (27–45) | 42 (28–50) | **<0.001** | 36 (25–46) |

*Abbreviations*: *SVR: sustained virological response, LSM: liver stiffness measurements, ALAT: alanine aminotransferase, pLSM: pretreatment liver stiffness measurement.*

Only 6.2% (29/467) of patients with ≥1 follow-up LSM had dLSM ≥0, while the majority (93.8) had a dLSM <0.

For patients without a follow-up LSM, the median pLSM was 9.9 kPa (IQR 6.0–17.1) compared to 12.4 kPa (IQR 7.4–18.8) among patients with a follow-up LSM. Among these, the median pLSM was 15.4 kPa (IQR 10.4–23.7) for patients with a dLSM ≥0 compared to 12.2 kPa (IQR 7.1–18) among those with a dLSM <0.

A dLSM ≥0 was significantly associated with incident decompensated cirrhosis (6.9% (2/29) vs 1.4% (6/438), p = 0.026) but not with HCC (3.5% (1/29) vs 1.8% (8/438), p = 0.538) or al-cause mortality (3.5% (1/29) vs 4.1% (18/438), p = 0.861).

## Hepatocellular carcinoma

Eleven (1.4%) patients developed HCC during the follow-up with an overall rate of HCC of 0.5/100 PY (95% CI 0.3–0.9/100 PY). The median follow-up for patients who developed HCC was 22.8 months (IQR 7.6–29.2) and the median pLSM was 27.0 kPa (IQR 17.5–45.0, range 4.7–69.1).

The median age at diagnosis was 60 years (IQR 59–63, range 53–69).

The optimized cut-off for incident HCC was 17.5 kPa which identified 9/11 of patients with incident HCC and had a sensitivity of 81.8% and specificity of 74.5% and a positive predictive value (PPV) and negative predictive value (NPV) of 4.4% and 99.7%, respectively (Table 2).

There was no significant difference in HCC incidence rate between patients with a pLSM <10 kPa and patients with a pLSM of 10–17.4 kPa (1/321 (0.31%), 0.12/100 PY vs 1/249 (0.4%), 0.14/100 PY, p = 0.925) but the HCC rate was significantly higher among patients with a pLSM ≥17.5 kPa (9/203 (4.43%), 1.40/100 PY, p = 0.017) (Table 3, Fig 2).

In comparison, a cut-off of 10 kPa identified 10/11 of patients with HCC and had a sensitivity and specificity of 58.5% and 90.9% and a PPV and NPV of 2.2% and 99.7%.

Of the two patients who were diagnosed with HCC post treatment, the first was a female in her mid-fifties with a pLSM of 4.7 kPa and no history of heavy alcohol use or diabetes while the other was a male in his late fifties with a pLSM of 13.0 kPa and a history of both heavy alcohol use and diabetes.

In univariate analysis, older age (sHR 1.08 (95% CI 1.00–1.11), p<0.001), diabetes (sHR 4.64 (95% CI 1.39–15.4), p = 0.012) and a pLSM ≥17.5 kPa (sHR 11.2 (95% CI 2.42–52.3), p = 0.002) was significantly associated with developing HCC post treatment (Table 4).

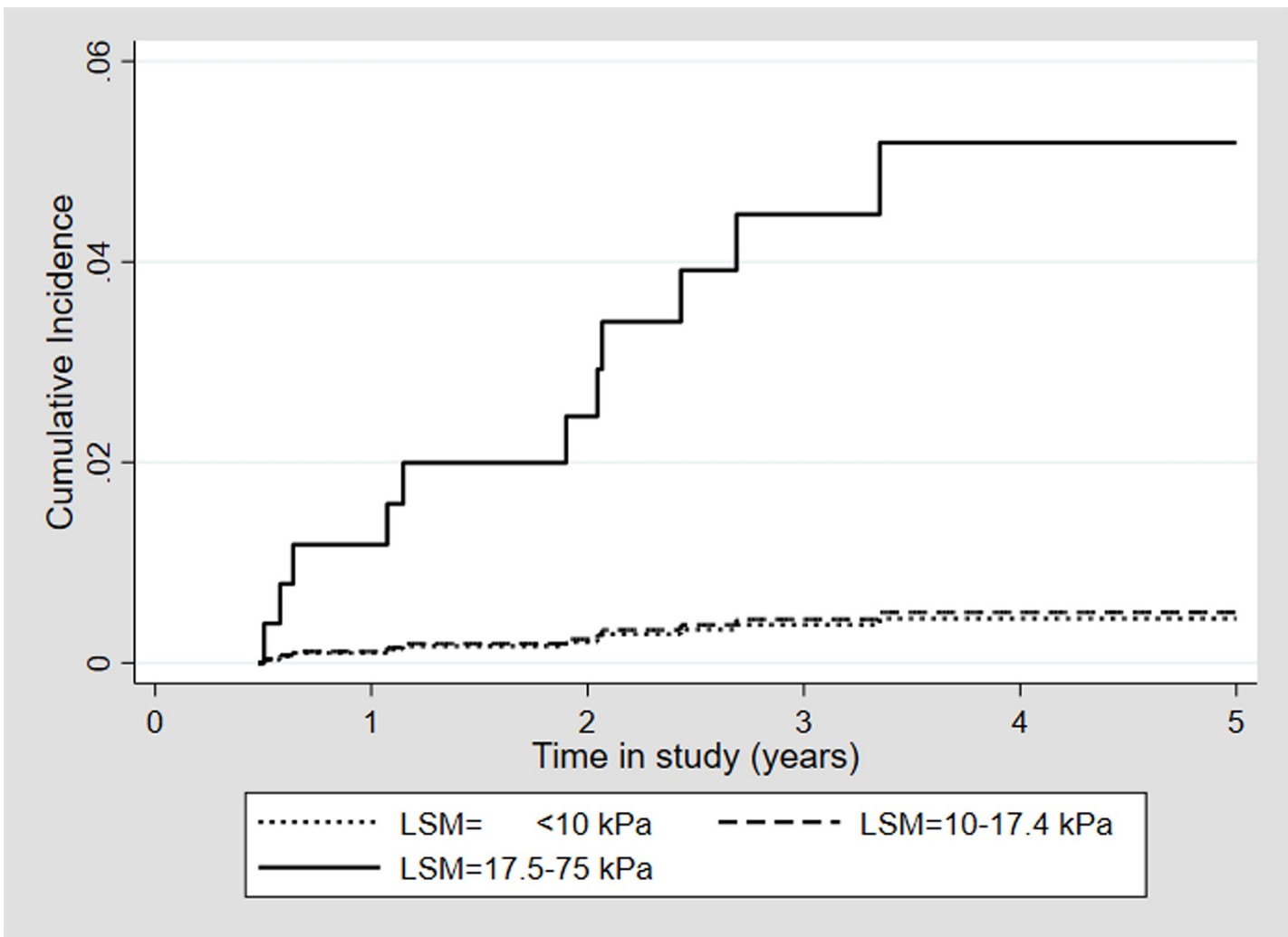

**Fig 2. Overall incidence of hepatocellular carcinoma for 773 patients achieving SVR after DAA treatment stratified by baseline LSM groups.** Abbreviations; SVR; sustained virological response, DAA; direct-acting antivirals, kPa; kilopascal, LSM; liver stiffness measurement.

In multivariate analysis only increasing age (sHR 1.07 (95% CI 1.03–1.12), p<0.001) and having a pLSM ≥17.5 kPa (HR 8.88 (95% CI 1.78–44.3), p = 0.003) was significantly associated with incident HCC while the association with diabetes (sHR 2.82 (95% CI 0.81–9.81), p = 0.103) did not reach statistical significance.

### Decompensated cirrhosis and all-cause mortality

Compensated cirrhosis was seen in 14 patients (1.8%) during follow-up, with an overall rate of 0.6/100 PY (95% CI 0.4–1.1) while 38 patients died (4.9%) yielding an all-cause mortality rate of 1.7/100 PY (95% CI 1.2–2.3/100 PY). The median follow-up for patients with incident decompensated cirrhosis was 7 months (IQR 2–13) and 26 months (IQR 15–30) among patients who died. The median pLSM was 38.8 kPa (IQR 26.3–56.1, range 12.3–70.6) among patients who developed decompensated cirrhosis during the follow-up period and 27.1 kPa (IQR 11.7–42.2, range 6.8–65.2) among patients who died.

**Table 2. Pretreatment LSM predicting decompensated cirrhosis, hepatocellular carcinoma and all-cause death during follow-up.**

|  | Cut-off (kPa) | Patients >Cut-off (%) | Sensitivity (%) | Specificity (%) | PPV (%) | NPV (%) |
|---|---|---|---|---|---|---|
| **Hepatocellular carcinoma** |  |  |  |  |  |  |
| Optimized cut-off | 17.5 | 26.3 | 81.8 | 74.5 | 4.4 | 99.7 |
| Cut-off for 90% sensitivity | 4.8 | 92.0 | 90.0 | 8.0 | 1.4 | 98.2 |
| Cut-off for 90% specificity | 28.0 | 9.8 | 45.5 | 90.0 | 6.2 | 99.1 |
| 12.5 kPa | 12.5 | 43.5 | 90.9 | 57.0 | 2.9 | 99.7 |
| 10 kPa | 10.0 | 58.5 | 90.9 | 42.0 | 2.2 | 99.7 |
| **Decompensated cirrhosis** |  |  |  |  |  |  |
| Optimized cut-off | 26.3 | 12.8 | 78.6 | 87.8 | 10.6 | 99.6 |
| Cut-off for 90% sensitivity | 12.3 | 45.3 | 90.0 | 56.1 | 3.6 | 99.7 |
| Cut-off for 90% specificity | 28.0 | 9.8 | 64.3 | 90.0 | 10.5 | 99.2 |
| 17.5 kPa | 17.5 | 26.3 | 78.6 | 74.7 | 5.4 | 99.5 |
| 12.5 kPa | 12.5 | 43.5 | 92.9 | 57.2 | 3.8 | 99.8 |
| 10 kPa | 10 | 58.5 | 100 | 42.3 | 3.1 | 100 |
| **All-cause death** |  |  |  |  |  |  |
| Optimized cut-off | 25.4 | 13.6 | 57.9 | 88.4 | 21.2 | 97.6 |
| Cut-off for 90% sensitivity | 6.9 | 71.9 | 90.0 | 27.6 | 6.0 | 98.2 |
| Cut-off for 90% specificity | 27.0 | 11.3 | 50.0 | 90.0 | 20.5 | 97.2 |
| 17.5 kPa | 17.5 | 26.3 | 63.2 | 75.7 | 11.8 | 97.6 |
| 12.5 kPa | 12.5 | 43.5 | 73.7 | 57.8 | 8.3 | 97.7 |
| 10 kPa | 10 | 58.5 | 81.6 | 42.7 | 6.8 | 97.8 |

*Abbreviations: kPa; kilopascal, PPV; positive predictive value, NPV; negative predictive value.*

During follow-up the median age at first episode of decompensated cirrhosis was 57 years (IQR 50–63, range 43–64) and the median age at the time of death was 59 years (IQR 53–64, range 40–67).

**Table 3. Incidence rates of cirrhosis decompensation, HCC and overall mortality according to pretreatment LSM for 773 patients with chronic hepatitis C treated with direct acting antivirals.**

|  | n per person-years | Incidence per 100 person year (95% CI) | Hazard ratio | p |
|---|---|---|---|---|
| **Hepatocellular carcinoma** |  |  |  |  |
| LSM <10 kPa | 1/851 | 0.12 (0.02–0.83) | 1 (reference) |  |
| LSM 10–17.4 kPa | 1/733 | 0.14 (0.02–0.97) | 1.14 (0.07–18.3) | 0.925 |
| LSM 17.5–75 kPa | 9/642 | 1.40 (0.73–2.70) | 12.3 (1.55–97.1) | 0.017 |
| **Decompensated cirrhosis** |  |  |  |  |
| LSM < 10 kPa | 0/851 | 0 (0–0.43)* | NA |  |
| LSM 10–17.4 kPa | 3/725 | 0.41 (0.13–1.30) | NA | 0.973 |
| LSM 17.5–75 kPa | 11/626 | 1.76 (1.0–3.17) | NA | <0.001 |
| **All-cause mortality** |  |  |  |  |
| LSM < 10 kPa | 7/851 | 0.82 (0.39–1.73) | 1 (reference) |  |
| LSM 10–17.4 kPa | 7/735 | 0.95 (0.45–2.0) | 1.19 (0.41–3.40) | 0.751 |
| LSM 17.5–75 kPa | 24/654 | 3.67 (2.46–5.47) | 4.39 (1.88–10.2) | 0.001 |

*Abbreviations: LSM; Liver stiffness measurement, kPa; kilopascal, CI; confidence interval, NA; not analyzed due to zero events in the reference group.*
*\* 0.975% CI.*

**Table 4. Factors associated with hepatocellular carcinoma, decompensated cirrhosis and all-cause mortality among 773 patients with chronic hepatitis C treated with direct acting antivirals.**

| | Hepatocellular carcinoma | | | |
| --- | --- | --- | --- | --- |
| | Univariate | | Multivariate | |
| Variable | sHR (95% CI) | p-value | sHR (95% CI) | p-value |
| Age, years | **1.08 (1.00–1.11)** | **<0.001** | 1.07 (1.03–1.12) | <0.001 |
| Male sex | 1.00 (0.29–3.44) | 0.992 | | |
| Western European | 0.83 (0.18–3.88) | 0.813 | | |
| ALAT at pLSM | 1.00 (0.99–1.01) | 0.556 | | |
| Diabetes | **4.64 (1.39–15.4)** | **0.012** | 2.82 (0.81–9.81) | 0.103 |
| Ever heavy alcohol use | 2.49 (0.66–9.37) | 0.178 | | |
| Ever intravenous drug use | 1.02 (0.30–3.50) | 0.975 | | |
| Days from pLSM to treatment | 1.00 (1.00.1.01) | 0.316 | | |
| Pretreatment LSM ≥17.5 kPa | **11.2 (2.42–52.5)** | **0.002** | **8.88 (1.78–44.3)** | **0.003** |
| Delta LSM ≥0 | 1.86 (0.23–15.2) | 0.560 | | |
| | Decompensated cirrhosis | | | |
| | Univariate | | Multivariate | |
| Variable | sHR (95% CI) | p-value | sHR (95% CI) | p-value |
| Age, years | 1.02 (0.98–1.06) | 0.328 | | |
| Male sex | 7.54 (0.99–57.2) | 0.051 | | |
| Western European | 1.12 (0.25–5.00) | 0.882 | | |
| ALAT at pLSM | 0.99 (0.98–1.01) | 0.297 | | |
| Diabetes | 1.42 (0.32–6.37) | 0.649 | | |
| Ever heavy alcohol use | 3.52 (0.98–12.6) | 0.053 | | |
| Ever intravenous drug use | 2.16 (0.60–7.72) | 0.237 | | |
| Days from pLSM to treatment | 1.00 (1.00–1.01) | 0.056 | | |
| Pretreatment LSM ≥17.5 kPa | **10.3 (2.87–36.7)** | **<0.001** | **10.3 (2.87–36.7)** | **<0.001** |
| Delta LSM ≥0 | **5.17 (1.06–25.2)** | **0.042** | | |
| | All-cause mortality | | | |
| | Univariate | | Multivariate | |
| Variable | HR (95% CI) | p-value | HR (95% CI) | p-value |
| Age, years | 1.03 (0.99–1.07) | 0.056 | | |
| Male sex | 1.60 (0.78–3.29) | 0.203 | | |
| Western European | 1.53 (0.54–4.30) | 0.425 | | |
| ALAT at pLSM | 0.99 (0.98–1.00) | 0.052 | | |
| Diabetes | **3.84 (1.94–7.61)** | **<0.001** | **3.01 (1.50–6.04)** | **0.002** |
| Ever heavy alcohol use | 1.75 (0.90–3.43) | 0.100 | | |
| Ever intravenous drug use | 1.29 (0.65–2.56) | 0.464 | | |
| Days from pLSM to treatment | 1.00 (0.99–1.01) | 0.483 | | |
| Pretreatment LSM ≥17.5 kPa | **4.05 (2.09–7.85)** | **<0.001** | **3.52 (1.80–6.90)** | **<0.001** |
| Delta LSM ≥0 | 0.91 (0.12–6.87) | 0.928 | | |

*Abbreviations*: *HR; hazard ratio, LSM; liver stiffness measurement, kPa; kilopascal.*

There was no difference among patients with a pLSM of <10 kPa and 10–17.4 kPa in incidence rates of decompensated cirrhosis (0/321 (0%), 0/100 PY vs 3/249 (1.2%), 0.41/100 PY, p = 0.925) (Table 3, Fig 3) or in all-cause mortality rates (7/321 (2.18%), 0.82/100 PY vs 7/249 (2.81%) 0.95/100 PY, p = 0.751) (Table 3, Fig 4) but patients with a pLSM ≥17.5 kPa had a significantly higher incidence rate of decompensated cirrhosis (11/203 (5.42%), 1.76/100 PY, p<0.001) and all-cause mortality rate (24/203 (11.8%), 3.67/100 PY, p = 0.001).

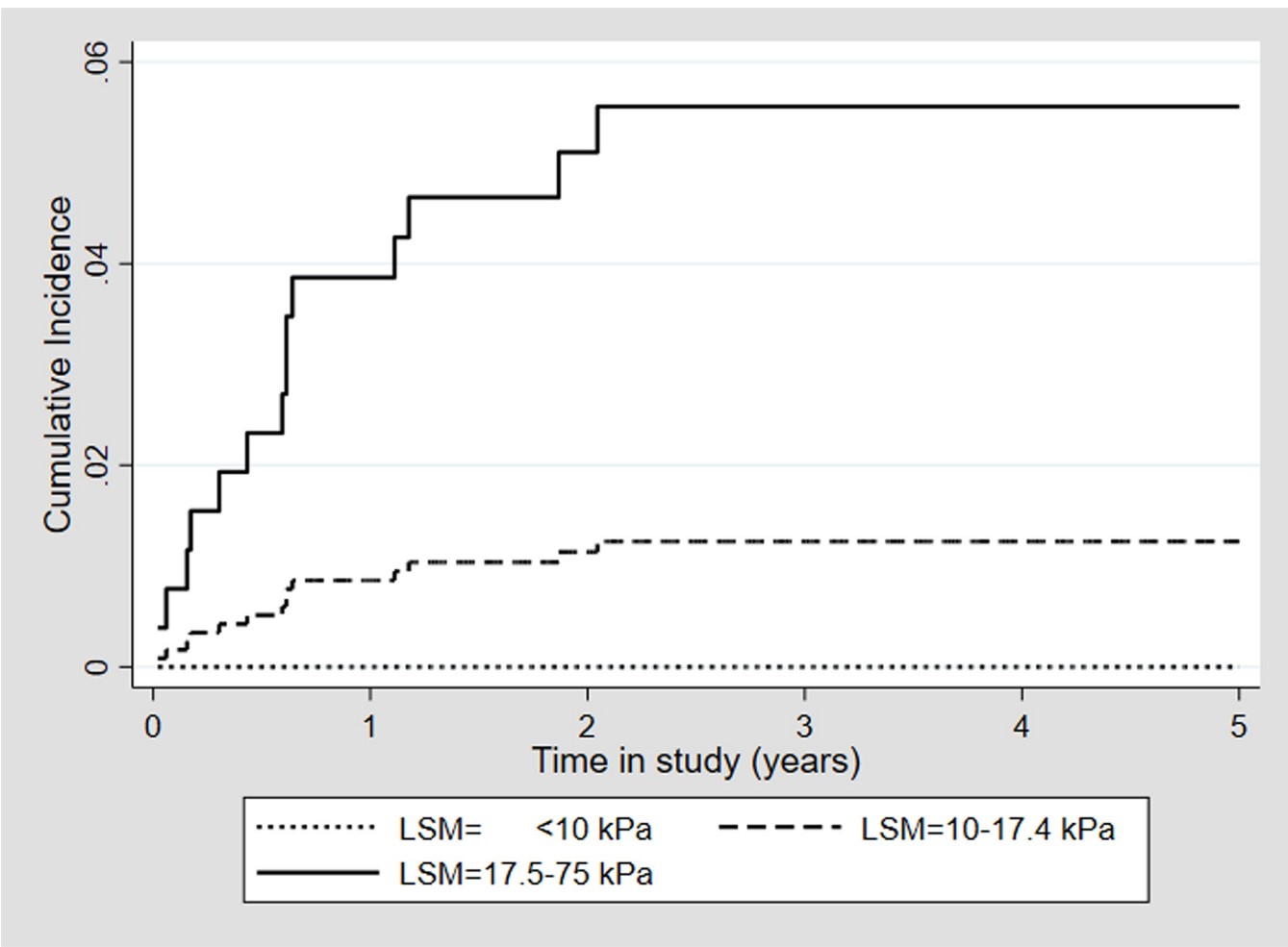

**Fig 3. Overall incidence of decompensated cirrhosis for 773 patients achieving SVR after DAA treatment stratified by baseline LSM groups.**
Abbreviations; SVR; sustained virological response, DAA; direct-acting antivirals, kPa; kilopascal, LSM; liver stiffness measurement.

The optimized cut-off for decompensated cirrhosis was 26.3 kPa, which identified 11/14 patients with incident decompensated cirrhosis with NPV of 99.7%, while the optimized cut-off for all-cause mortality was 25.4 kPa, which identified 22/38 of patients who died during the follow-up and NPV of 97.6% (Table 2).

A cut-off of 17.5 kPa identified 11/14 of patients with decompensated cirrhosis with NPV 99.5%, while it identified 24/38 of patients who died of all causes during follow-up with NPV of 97.6% (Table 2).

In univariate analysis, a pLSM ≥17.5 kPa (sHR 10.3 (95% CI 2.87–36.7)), p<0.001) and a dLSM ≥0 (sHR 5.17 (95% CI 1.06–25.2)) was significantly associated with decompensated cirrhosis (Table 4).

All-cause mortality was significantly associated with diabetes in both univariate (HR 3.84 (95% CI 1.44–5.78, p<0.001) and multivariate analysis (HR 3.01 (95% CI 1.50–6.04) p = 0.002) and with having a pLSM≥17.5 kPa in univariate (HR 4.05 (95% CI 2.09–7.85), p<0.001) and multivariate analysis (HR 3.52 (95% CI 1.80–6.90), p<0.001) (Table 4).

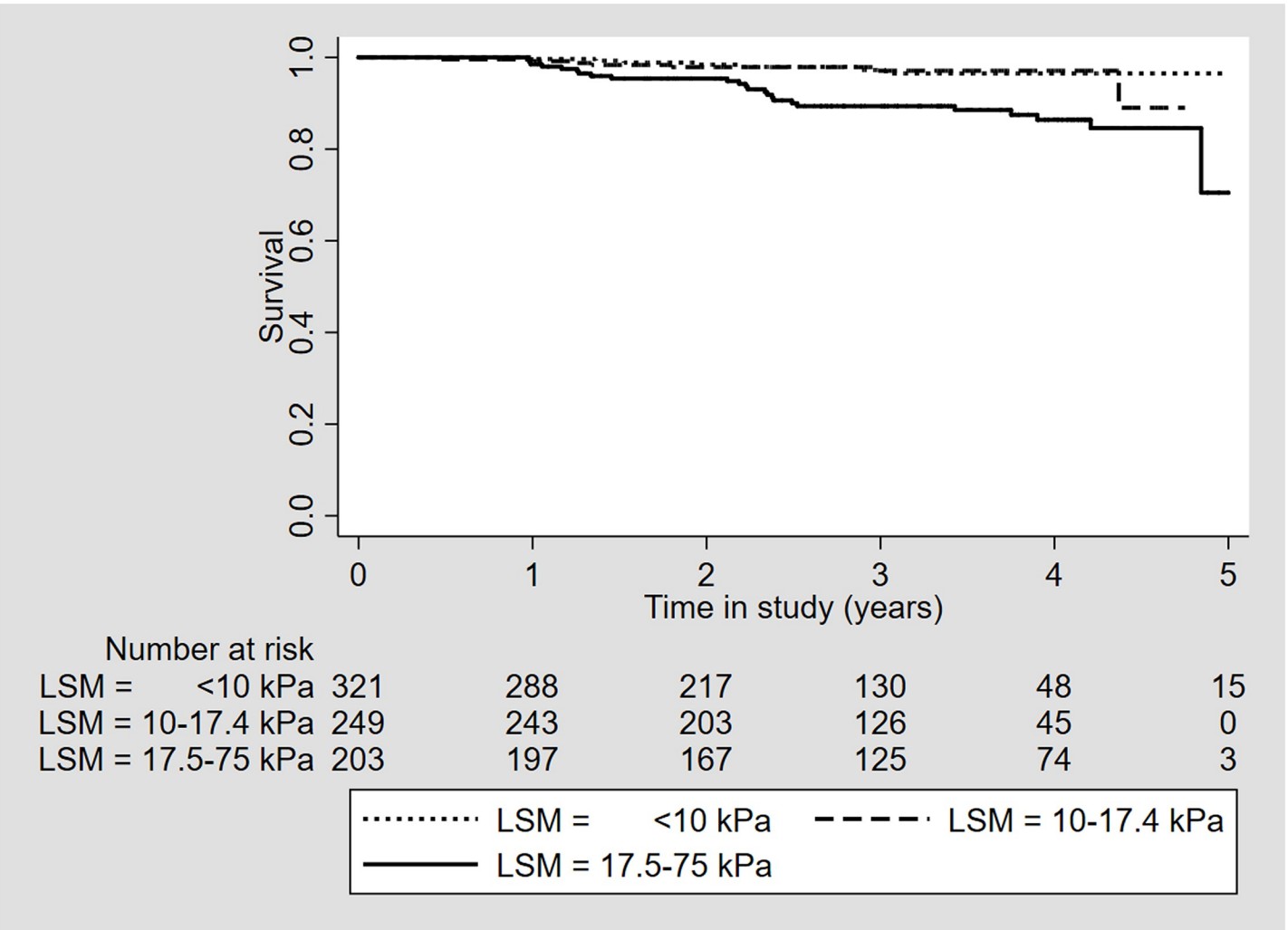

**Fig 4. Overall survival for 773 patients achieving SVR after DAA treatment stratified by baseline LSM group.** Abbreviations; SVR; sustained virological response, DAA; direct-acting antivirals, kPa; kilopascal, LSM; liver stiffness measurement.

## Discussion

The findings of this Danish nationwide cohort study showed that pLSM has a prognostic value for the development of HCC, decompensated cirrhosis, and overall survival in CHC patients achieving SVR after DAA therapy. We found a cut-off of 17.5 kPa to be a good predictor of incident HCC with incidence rates increasing 10-fold above this cut-off compared to patients with a pLSM below 17.5 kPa. We did not find a significantly higher risk of developing HCC in patients with a pLSM of 10–17.4 kPa compared to patients with a pLSM <10 kPa. This was unexpected and may have been due to the low number of patients with HCC in our study or the length of the follow-up. The one patient with a pLSM of 10–17.4 kPa who developed HCC during follow-up had multiple known risk factors for HCC in the form of older age, male sex, alcohol abuse and diabetes. As the negative predictive value for the cut-off of 17.5 kPa was very high at 99.7%, this could suggest that within the first three years after cure for hepatitis C, monoinfected patients with no prior episode of HCC or decompensated cirrhosis and with a pretreatment LSM below 17.5 kPa may not benefit from HCC surveillance. However, patients

with a pLSM of 10–17.4 kPa and other risk factors for HCC should also be considered for HCC screening, based on individual risk assessment.

Our findings are in line with recent study by Shiha et al. [31] that followed 2372 patients monoinfected with HCV with no prior episodes of decompensated cirrhosis or HCC who achieved SVR after treatment with DAA for an average of 23.6 months. In 638 patients with a pretreatment LSM of 10.3–16.3 kPa Shiba et al. found a HCC incidence rate of 0.664/100 PY compared to 2.917/100 PY among 1734 patients with a pretreatment LSM of ≥16.4 kPa.

Also, in a study including 572 CHC patients treated with DAA and pretreatment LSM ≥10 kPa and no prior episodes of decompensated cirrhosis of HCC, Pons et al. [16] found a greater proportion of patients who developed HCC among patients with a pretreatment LSM of ≥20 kPa compared to 10–19.9 kPa (6.1% (13/212) vs 3.6% (12/360), p = 0.114) during a median follow-up of 2.8 years. Among the five patients in the study who developed decompensated cirrhosis, all had a pretreatment LSM ≥20 kPa.

Similarly, Hansen et al. [32] followed 591 patients with chronic hepatitis C for a median of 46.1 months and found that cirrhotic complications, defined as first episode of HCC or decompensated cirrhosis, occurred almost exclusively in patients with a baseline LSM of 17 kPa, with a negative predictive value of 98.0%.

A recent paper estimated that biannual screening for HCC in patients with SVR after treatment for CHC would be cost effective in patients with an HCC incidence of ≥1.32% per year with an incremental cost-effectiveness ratio (ICER) < $50,000/quality adjusted life-year (QALY) [33]. Also, apart from cost effectiveness, screening patients with low incidence rates increases the risk of having a false positive ultrasound-based HCC diagnosis [33, 34], which leads to additional tests, in some cases including biopsy, perhaps therapy and the anxiety of a cancer diagnosis [35]. The results from our study, as well as the study by Shiha et al. [31], suggest that patients with a pLSM of <17.5 kPa can be safely omitted from post treatment HCC screening. However further studies are needed to clarify if this holds true during prolonged and longer follow-up.

A drop in LSM after treatment was associated with a reduction in incident decompensated cirrhosis but not incident HCC or overall-death. A recent study by Pons et al. of CHC patients, treated with DAA and pLSM ≥10 kPa, also showed no statistically significant association between having a decrease of ≥20% in LSM at one-year follow-up after treatment and reduction in HCC incidence [16]. Conversely, Ravaioli et al. found that a reduction in LSM of >30% from baseline to end of treatment with DAA for CHC was inversely associated with development of HCC in a smaller, retrospective study with 139 patients with Child-Pugh A and B cirrhosis [19].

The low number of incident HCC among patients with a follow-up LSM in our study would make it difficult to show a difference in outcome among the groups with delta LSM ≥0 and <0, respectively.

This study has several limitations. We derived our data retrospectively from registers which might be expected to result in less accurate ascertainment of exposures and outcomes than a prospective follow-up. Secondly, the pLSM was not performed on the date of treatment initiation in most patients and could have changed in either direction in the intervening time [36]. However, in most of the patients, the pLSM was performed less than six months before treatment initiation and time from pLSM to index date was not significantly associated with outcomes in the regression analyses. Thirdly, we did not have data on whether pLSM were performed with the patients fasting. As LSM can be falsely elevated if the patient is not fasting [37] this could have caused an overestimation of the LSM cutoffs in the study.

However, patients in Denmark are instructed to be fasting when having LSM performed and as almost all patients had at least one LSM prior to the pLSM, most would be expected to have been fasting at the time of the pLSM.

Also, the follow-up time in our study was rather short (36 months). However, a recent study by Iounnaou et al. [4] showed that the elevated risk of HCC after SVR in cirrhotic patients treated for CHC with interferon based or interferon free therapy did not increase during a follow-up of up to 10 years, but further long-term studies are needed to confirm this.

It would have been a great advantage to the study if serological markers of liver fibrosis, such as FIB-4, had been available to corroborate our LSM findings, but as aspartate aminotransferase (AST) was not a standard test in Denmark during the study period, and only available for a small proportion of patients.

It would have been preferable to also have LSM at end of treatment (EOT) or at SVR as the inflammation caused by CHC can cause an elevation of LSM, regardless of fibrosis and LSM at SVR could be more accurate at predicting outcomes after DAA treatment [38]. However, LSM at EOT or SVR was not available in most patients. Furthermore, a significant part of patients in treatment for CHC are lost to follow up, especially vulnerable patients like those with active injecting drug use or suffering from homelessness [39]. Being able to provide prognostication and reassurance at the time of treatment initiation would be important, especially in patients at risk of lost to follow up after treatment.

## Conclusions

In conclusion, our study suggests that pLSM can be used to risk stratify CHC patients with no previous episode of decompensated cirrhosis or HCC who achieve SVR after treatment with DAAs. Patients in this group with a pretreatment LSM below 17.5 kPa without other risk factors for cirrhosis and HCC appear not to benefit from HCC surveillance within the first 3 years of cure. However, further longtime follow-up studies are needed to confirm our findings and to address whether HCC screening can be avoided hereafter.

## Supporting information

**S1 Appendix. Definition of specific diagnoses.**
(DOCX)

## Author Contributions

**Conceptualization:** Jacob Søholm, Janne Fuglsang Hansen, Belinda Mössner, Alex Lauersen, Nina Weis, Peer Brehm Christensen.

**Data curation:** Jacob Søholm, Belinda Mössner, Birgit Thorup Røge, Alex Lauersen, Jesper Bach Hansen, Nina Weis, Toke Seierøe Barfod, Suzanne Lunding, Anne Øvrehus, Rajesh Mohey, Peter Thielsen, Peer Brehm Christensen.

**Formal analysis:** Jacob Søholm, Janne Fuglsang Hansen, Belinda Mössner, Peer Brehm Christensen.

**Funding acquisition:** Jacob Søholm, Peer Brehm Christensen.

**Methodology:** Jacob Søholm, Janne Fuglsang Hansen, Belinda Mössner, Birgit Thorup Røge, Alex Lauersen, Jesper Bach Hansen, Nina Weis, Toke Seierøe Barfod, Suzanne Lunding, Peer Brehm Christensen.

**Project administration:** Jacob Søholm.

**Supervision:** Peer Brehm Christensen.

**Visualization:** Jacob Søholm, Anne Øvrehus.

**Writing – original draft:** Jacob Søholm, Jesper Bach Hansen.

**Writing – review & editing:** Janne Fuglsang Hansen, Belinda Mössner, Birgit Thorup Røge, Alex Lauersen, Nina Weis, Toke Seierøe Barfod, Suzanne Lunding, Anne Øvrehus, Rajesh Mohey, Peter Thielsen, Peer Brehm Christensen.

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
