## [Decision Letter · Decision Letter 0]

19 Aug 2020

PONE-D-20-23907

Low incidence of HCC in chronic hepatitis C patients with pretreatment liver stiffness measurements below 17.5 kilopascal who achieve SVR following DAAs

PLOS ONE

Dear Dr. Søholm,

Thank you for submitting your manuscript to PLOS ONE. After careful consideration, we feel that it has merit but does not fully meet PLOS ONE’s publication criteria as it currently stands. Therefore, we invite you to submit a revised version of the manuscript that addresses the points raised during the review process.

As you can see, both reviewers appreciated your work, but also raised a couple of important issues. Most importantly, the low number of events greatly limits the power of the study and this limitation needs to be clearly highlighted. Along the same lines, some of the conclusions should probably be softened a bit.

We look forward to receiving your revised manuscript.

Kind regards,

Pavel Strnad

Academic Editor

PLOS ONE

Journal Requirements:

2.We note that you have indicated that data from this study are available upon request. PLOS only allows data to be available upon request if there are legal or ethical restrictions on sharing data publicly. For information on unacceptable data access restrictions, please see http://journals.plos.org/plosone/s/data-availability#loc-unacceptable-data-access-restrictions.

Reviewers' comments:

Reviewer's Responses to Questions

**Comments to the Author**

1. Is the manuscript technically sound, and do the data support the conclusions?

Reviewer #1: Yes

Reviewer #2: Yes

2. Has the statistical analysis been performed appropriately and rigorously? 

Reviewer #1: I Don't Know

Reviewer #2: Yes

3. Have the authors made all data underlying the findings in their manuscript fully available?

Reviewer #1: Yes

Reviewer #2: Yes

4. Is the manuscript presented in an intelligible fashion and written in standard English?

Reviewer #1: Yes

Reviewer #2: Yes

5. Review Comments to the Author

Reviewer #1: Jacob Søholm et al. Investigated if liver stiffness measurement (LSM) could predict the risk of HCC, decompensation and all-cause mortality in patients with SVR after DAA treatment.

This is a retrospective study of 773 patients.

The authors suggest that Patients after HCV cure by DAA therapy with a pretreatment LSM <17.5 kPa appear not to benefit from HCC surveillance for the first 3 years after treatment.

Several other studies have performed similar studies.

Comments:

„We did not find a significantly higher risk of developing HCC in patients with a pLSM of 10-17.4 kPa compared to patients with a pLSM <10 kPa“

LSM is influenced by several factors, e.g. it has been shown that values can be higher in patients that have had a meal withing the last 2 hours prior to the investigation. This needs to be considered.

In addition, pretreatment LSM can be influenced by the ALT value (inflammation). Ideally, a cut-off at the end of the treatment could be more accurate? (i.e. Clin Infect Dis. 2019 Nov 22;ciz1140.)

Only 11 patients in that cohort developed HCC. 9/11 patients could be identified by the cut-off of 17.5 kPa. The authors mentioned that LSM >17.5 kPA was associated with age, male sex and diabetes. All these factors have been associated with HCC risk. Were the other 2 patients male, had older age and diabetes? Are these factors (in combination) more relevant than LSM? I wonder if the authors would not perform HCC surveillance in a male patient, with LSM of 14 kPA and diabetes mellitus? Thus, in my view the concluison is too stroing in my view based on the 11 patients with HCC.

LSM may not be available in all settings. Other easier to use parameter may be more valuable.

Several studies have investigated albumin (J Hepatol. 2020 Mar;72(3):472-480) or FIB-4 (e.g. Gastroenterology. 2019 Nov;157(5):1264-1278.e4.) as predictive marker. FIB-4 data could improve the study.

Reviewer #2: Soholm et al performed a retrospective analysis of about 800 patients to predict HCC, complications and mortality depending on the liver stiffness in patients after successful (SVR) DAA therapy. They could identify 17.5 kPA as LSM cut-off with a NPV for any complication >95% suggesting that in those patients closed surveillance after SVR is not required. Although novelty is limited, data analysis and the study design are of good quality.

I have just some minor comments:

A valid LSM <2 years before treatment with DAA which is the index day might be quite long and might include a selection bias – subgroup analysis or time between LSM and index date as confounder in the multivariate would be important to show

As the primary outcome was HCC but more patient died than developing HCC during follow, authors should state and discuss how they dealt with competing risk (death before HCC). The figures show cumulative incidences which suggest that this was considered during analysis.

Multivariate data might be over fitted as incidence of 11 (HCC), 14 (dec. cirrhosis) and 38 for death restrict the number of confounder to maximum of four when calculating the risk of death and even 1-2 for the other endpoints.

6. PLOS authors have the option to publish the peer review history of their article (what does this mean?). If published, this will include your full peer review and any attached files.

Reviewer #1: No

Reviewer #2: No

---

## [Author Response · Author response to Decision Letter 0]

8 Oct 2020

Response to Reviewers

We would like to thank the academic editor and the reviewers for their comments.

We have addressed the queries in order below

Response to the academic editor

1: Editor’s comment: Please ensure that your manuscript meets PLOS ONE's style requirements, including those for file naming. 

Author’s reply: We have tried to adhere to the PLOS ONE’s style requirements. 

2: Editor’s comment: We note that you have indicated that data from this study are available upon request. PLOS only allows data to be available upon request if there are legal or ethical restrictions on sharing data publicly.

Author’s reply: We have addressed the issue of data access in the cover letter. 

Response to reviewer 1:

1: reviewer’s comment: „We did not find a significantly higher risk of developing HCC in patients with a pLSM of 10-17.4 kPa compared to patients with a pLSM <10 kPa“.

LSM is influenced by several factors, e.g. it has been shown that values can be higher in patients that have had a meal withing the last 2 hours prior to the investigation. This needs to be considered.

In addition, pretreatment LSM can be influenced by the ALT value (inflammation). Ideally, a cut-off at the end of the treatment could be more accurate? (i.e. Clin Infect Dis. 2019 Nov 22;ciz1140.)

Authors’s reply: We have discussed the first point in the discussion:

Thirdly, we did not have data on whether pLSM were performed with the patients fasting. As LSM can be falsely elevated if the patient is not fasting this could have caused an overestimation of the LSM cutoffs in the study.

However, patients in Denmark are instructed to be fasting when having LSM performed and as almost all patients had at least one LSM prior to the pLSM, most would be expected to have been fasting at the time of the pLSM.

We also agree that also having a LSM at end of treatment or SVR would have been preferable, but unfortunately, it was not available in most patients. We have added the following to the discussion: It would have been preferable to also have LSM at end of treatment (EOT) or at SVR as the inflammation caused by CHC can cause an elevation of LSM, regardless of fibrosis and LSM at SVR could be more accurate at predicting outcomes after DAA treatment [38]. However, LSM at EOT or SVR was not available in most patients. Furthermore, a significant part of patients in treatment for CHC are lost to follow up, especially vulnerable patients like those with active injecting drug use or suffering from homelessness [39]. Being able to provide prognostication and reassurance at the time of treatment initiation would be important, especially in patients at risk of lost to follow up after treatment.

 2: reviewer’s comment: “Only 11 patients in that cohort developed HCC. 9/11 patients could be identified by the cut-off of 17.5 kPa. The authors mentioned that LSM >17.5 kPA was associated with age, male sex and diabetes. All these factors have been associated with HCC risk. Were the other 2 patients male, had older age and diabetes? Are these factors (in combination) more relevant than LSM? I wonder if the authors would not perform HCC surveillance in a male patient, with LSM of 14 kPA and diabetes mellitus? Thus, in my view the concluison is too stroing in my view based on the 11 patients with HCC.”

Author’s reply:

We have described the two patients with a LSM <17.4 under the heading “Hepatocellular carcinoma”: Of the two patients who were diagnosed with HCC post treatment, the first was a female in her mid-fifties with a pLSM of 4.7 kPa and no history of heavy alcohol use or diabetes while the other was a male in his late fifties with a pLSM of 13.0 kPa and a history of both heavy alcohol use and diabetes.

We also added the following sentence in the discussion:

“As the negative predictive value for the cut-off of 17.5 kPa was very high at 99.7 %, this could suggest that within the first three years after cure for hepatitis C, monoinfected patients with no prior episode of HCC or decompensated cirrhosis and with a pretreatment LSM below 17.5 kPa may not benefit from HCC surveillance. However, patients with a pLSM of 10-17.4 kPa and other risk factors for HCC should also be considered for HCC screening, based on individual risk assessment.”

3: reviewer’s comment: LSM may not be available in all settings. Other easier to use parameter may be more valuable.

Several studies have investigated albumin (J Hepatol. 2020 Mar;72(3):472-480) or FIB-4 (e.g. Gastroenterology. 2019 Nov;157(5):1264-1278.e4.) as predictive marker. FIB-4 data could improve the study.

Author’s reply: We agree that adding biomarkers would have improved the study. Unfortunately,blood samples, including albumin and especially ASAT were only available for a minority of patients in the study and we therefore could not include albumin or FIB-4 in the analyses.

We have mentioned this in discussion: It would have been a great advantage to the study if serological markers of liver fibrosis, such as FIB-4, had been available to corroborate our LSM findings, but as aspartate aminotransferase (AST) was not a standard test in Denmark during the study period, and only available for a small proportion of patients. 

We also agree that the use of LSM is not available in all settings, limiting the applicability of the study, but when available, it can be practical in outreach programs among the most vulnerable patients where one contact prior to treatment initiation (using LSM and POC HCV RNA testing) is preferable. 

We have added the following text in the introduction: Using LSM as a predicting marker allows for less contacts with health care providers before treatment initiation, as compared to biomarkers. This can be advantageous in outreach programs among marginalized populations, such as homeless people and people who inject drugs.

Response to reviewer 2: 

1: reviewer’s comment: A valid LSM <2 years before treatment with DAA which is the index day might be quite long and might include a selection bias – subgroup analysis or time between LSM and index date as confounder in the multivariate would be important to show.

Author’s reply: We agree that the long time from LSM to index date in some patients could represent a bias and have included the time between LSM and index date as a variable in the regression analyses. 

2: reviewer’s comment: the primary outcome was HCC but more patient died than developing HCC during follow, authors should state and discuss how they dealt with competing risk (death before HCC). The figures show cumulative incidences which suggest that this was considered during analysis.

Author’s reply: We agree that competing risk analysis should be used for HCC and decompensated cirrhosis and have changed from cox regression to competing risk regression and reported subhazard ratios for these outcomes.

3: reviewer’s comment: Multivariate data might be over fitted as incidence of 11 (HCC), 14 (dec. cirrhosis) and 38 for death restrict the number of confounder to maximum of four when calculating the risk of death and even 1-2 for the other endpoints.

Authors reply: We have lowered the cutoff for including variables in multivariate analyzes from p<0.1 to p<0.05, thus lowering the number of confounders included.

---

## [Decision Letter · Decision Letter 1]

30 Oct 2020

PONE-D-20-23907R1

Low incidence of HCC in chronic hepatitis C patients with pretreatment liver stiffness measurements below 17.5 kilopascal who achieve SVR following DAAs

PLOS ONE

Dear Dr. Søholm,

Thank you for submitting your manuscript to PLOS ONE. After careful consideration, we feel that it has merit but does not fully meet PLOS ONE’s publication criteria as it currently stands. Therefore, we invite you to submit a revised version of the manuscript that addresses the points raised during the review process.

As you can see, the reviewers aggreed that the manuscript substantially improved and only a minor revision is needed at this step.

We look forward to receiving your revised manuscript.

Kind regards,

Pavel Strnad

Academic Editor

PLOS ONE

Reviewers' comments:

Reviewer's Responses to Questions

**Comments to the Author**

1. If the authors have adequately addressed your comments raised in a previous round of review and you feel that this manuscript is now acceptable for publication, you may indicate that here to bypass the “Comments to the Author” section, enter your conflict of interest statement in the “Confidential to Editor” section, and submit your "Accept" recommendation.

Reviewer #1: All comments have been addressed

Reviewer #2: (No Response)

2. Is the manuscript technically sound, and do the data support the conclusions?

Reviewer #1: Yes

Reviewer #2: Yes

3. Has the statistical analysis been performed appropriately and rigorously? 

Reviewer #1: Yes

Reviewer #2: Yes

4. Have the authors made all data underlying the findings in their manuscript fully available?

Reviewer #1: (No Response)

Reviewer #2: No

5. Is the manuscript presented in an intelligible fashion and written in standard English?

Reviewer #1: (No Response)

Reviewer #2: Yes

6. Review Comments to the Author

Reviewer #1: The authors have taken up my comments and critically discussed the limitations of the study. No further comments

Reviewer #2: Soholm et al. addressed my previous comments and improved the impact of the manuscript.

I have only one minor comment regarding the competing risk analysis: Authors should mention in the stats section or results section why this analysis was necessary - as death was competing substantially with the occurence of HCC or complications of cirrhosis.

7. PLOS authors have the option to publish the peer review history of their article (what does this mean?). If published, this will include your full peer review and any attached files.

Reviewer #1: No

Reviewer #2: No

---

## [Author Response · Author response to Decision Letter 1]

27 Nov 2020

Response to Reviewers

We would like to thank the the reviewers for their comments.

We have addressed the queries in order below

Response to reviewer 2: 

1: reviewer’s comment: Authors should mention in the stats section or results section why this analysis was necessary - as death was competing substantially with the occurence of HCC or complications of cirrhosis.

Author’s reply: We agree that this should be specified and have rewritten the section in the statistics section:

Cox regression was used to estimate hazard ratios for all-cause death. Competing risk regression was used to estimate subhazard ratios for HCC and decompensated cirrhosis, as death was a substantial competing risk for these two outcomes.

---

## [Editor Report · Decision Letter 2]

30 Nov 2020

Low incidence of HCC in chronic hepatitis C patients with pretreatment liver stiffness measurements below 17.5 kilopascal who achieve SVR following DAAs

PONE-D-20-23907R2

Dear Dr. Søholm,

We’re pleased to inform you that your manuscript has been judged scientifically suitable for publication and will be formally accepted for publication once it meets all outstanding technical requirements.

Kind regards,

Pavel Strnad

Academic Editor

PLOS ONE
---

## [Editor Report · Acceptance letter]

2 Dec 2020

PONE-D-20-23907R2 

Low incidence of HCC in chronic hepatitis C patients with pretreatment liver stiffness measurements below 17.5 kilopascal who achieve SVR following DAAs 

Dear Dr. Søholm:

I'm pleased to inform you that your manuscript has been deemed suitable for publication in PLOS ONE. Congratulations! Your manuscript is now with our production department. 

Kind regards, 

on behalf of

Dr. Pavel Strnad 

Academic Editor

PLOS ONE